# Estimating Infection-Related Human Mobility Networks Based on Time Series Data of COVID-19 Infection in Japan

**Tetsuya Yamada** [1,*,†] and **Shoi Shi** [2,*]

1    Faculty of Medicine, The University of Tokyo, Tokyo 113-0033, Japan
2    International Institute for Integrative Sleep Medicine (WPI-IIIS), University of Tsukuba, Tsukuba 305-8575, Japan
*    Correspondence: tetsuyamada1222@gmail.com (T.Y.); shoishi0322@gmail.com (S.S.)
†    Current address: Center for Molecular Biology (ZMBH), DKFZ-ZMBH Alliance, Heidelberg University, 69120 Heidelberg, Germany.

**Abstract:** *Background:* Comprehensive and evidence-based countermeasures against emerging infectious diseases have become increasingly important in recent years. COVID-19 and many other infectious diseases are spread by human movement and contact, but complex transportation networks in the 21st century make it difficult to predict disease spread in rapidly changing situations. It is especially challenging to estimate the network of infection transmission in countries where traffic and human movement data infrastructure is not yet developed. *Methods:* In this study, we devised a method utilizing an ordinary and partial differential equations-based mathematical model and a modified mathematical optimization method to estimate the network of transmission of COVID-19 from the time series data of its infection and applied it to determine its spread across areas in Japan. Furthermore, utilizing the estimated human mobility network, we predicted the spread of infection using the Tokyo Olympics as a model. *Findings:* We incorporated the effects of soft lockdowns, such as the declaration of a state of emergency, and changes in the infection network due to government-sponsored travel promotion, and revealed that the estimated effective distance captured human mobility changing dynamically in the different stages of the pandemic. The model predicted that the Tokyo Olympic and Paralympic Games would increase the number of infected cases in the host prefectures by up to 80%. *Interpretation:* The models used in this study are available online, and our data-driven infection network models are scalable, whether it be at the level of a city, town, country, or continent, and applicable anywhere in the world, as long as the time-series data of infections per region is available. These estimations of effective distance and the depiction of infectious disease networks based on actual infection data are expected to be useful in devising data-driven countermeasures against emerging infectious diseases worldwide.

**Keywords:** COVID-19; Japan; state of emergency; SEIR model; diffusion equation; MCMC





## 1. Research in Context

### 1.1. Evidence before This Study

The situation with the coronavirus disease 2019 (COVID-19) is constantly changing due to various interventions such as lockdowns, states of emergency (i.e., soft lockdowns), travel encouragements or restrictions, and vaccination or the emergence of new variants of the virus. While lockdown has been proven to be one of the most effective infection control measures, it severely slows down the economy. Thus, it would be essential to develop policies that balance both economic activity and medical resources in order to adapt to a new normal during the COVID-19 pandemic. Because the spread of infection between cities is caused by the movement of people, estimating human mobility networks would be a promising approach to predicting the spread of COVID-19 and taking effective response measures against the pandemic. However, existing schemes require human movement and

traffic data extensively, which makes it difficult to be adopted in countries where the data infrastructure is not yet developed adequately. Considering the importance of tackling the COVID-19 pandemic across the globe, without severe restrictions on economic activity, it is required to develop a scheme for estimating infection-related human mobility networks based on the minimum necessary information.

### 1.2. Added Value of This Study

This study proposes a new scheme to estimate the dynamically changing human mobility networks based on time-series data of the daily number of COVID-19 infections in each prefecture in Japan, the host country of the Tokyo 2020 Olympic and Paralympic Games. The model, based on the "Susceptible-Exposed-Infectious-Recovered" (SEIR) model and the diffusion equation, recapitulated the number of infected people in each prefecture, and effective distances between prefectures were estimated. The estimated effective distances dynamically changed at the different stages of the pandemic, which reflected the changes in human movement in response to the government policies such as the state of emergency and travel campaigns. In addition, a modified mathematical optimization method was implemented to visualize the infection-related human mobility networks in the host country, revealing hidden patterns of the spread of infection. Lastly, utilizing the estimated human mobility network, we quantitatively estimated the effect of the Olympic and Paralympic Games on the pandemic in the host country. The prediction based on the model proposed effective and safe ways to hold the games while preventing the severe increase in infection in each prefecture.

### 1.3. Implications of All the Available Evidence

Japan is a unique country where a legally binding lockdown has not been declared, but various policies (e.g., state of emergency and travel promoting campaign) that have a great impact on human mobility have been developed. Therefore, we consider that the dynamics of COVID-19 infection in Japan is a suitable model for estimating infection-related human mobility networks and providing a new strategy to contain the pandemic based on real-life-based networks. In the present study, the infection-related human mobility networks in the host country were estimated based on daily infected cases and relative geographical relationships. The network made it possible to visualize the spread of the virus across the country at different stages of the pandemic. This time- and space-resolved information could be valuable to implement fine-grained response measures that can balance economic activities and limited medical resources. Furthermore, due to the simplicity of the scheme, which does not require an extensive data infrastructure, it can be applied to most parts of the world. This new scheme estimating human mobility networks based on the minimum necessary information would be helpful to develop proper and effective measures against the pandemic across the world, which can ultimately lead to the prevention of the emergence of new variants and overcome the pandemic.

## 2. Introduction

Since the identification of the coronavirus disease 2019 (COVID-19) in December 2019 [1,2], over 180 million laboratory-confirmed infections and 4 million deaths have been observed worldwide as of 30 June 2021 [3]. The development of vaccines has certainly reduced the number of new infections worldwide [4,5], allowing life before COVID-19 to gradually return to normal. However, in countries where vaccination has been delayed, there is a risk that the infection will continue to spread, and the emergence of new mutant virus strains has been observed [6–8]; thus, proper measures against COVID-19 are still required.

In the past year and a half, effective countermeasures against COVID-19 have been established, such as social distancing and wearing face masks. The effectiveness of these non-pharmaceutical interventions (NPIs) has also been quantitatively evaluated [9–12], together with the need for appropriate application where necessary depending on the active status of COVID-19. Otherwise, governments have to take painful and severe

measures again, such as lockdowns, to prevent the spread of infection. Lockdowns are very effective infection control measures [10,13,14]; however, the trade-off between the impact on the economy and on the mental health of individuals cannot be ignored [15–18]; thus, large-scale lockdown should be avoided as much as possible. As an alternative, selective lockdowns, which prevent the spread of infection by locking down a city when the number of infected individuals increases in that city, or other specific interventions have also been applied with some success in the United Kingdom and in other countries [19–23].

Since the spread of infection between cities is caused by the movement of people [24–26], the infection does not always spread to cities that neighbor geographically but spreads between cities where there is a continuous flow of individuals [27,28]. Therefore, it is useful to create a network for predicting the spread of infection by calculating effective distances based on traffic and mobility of individuals [29,30]. In fact, this method has been successful in predicting the spread of infection between countries and cities [31]. Conversely, effective distances are easier to estimate in countries where human movement and traffic data are easy to be obtained and are well developed, although, this is limited to a few countries. Even in the countries or regions where the human movement and traffic data infrastructure is not yet developed, it is required to share a scheme for estimating infection networks based on the minimum necessary information and taking appropriate countermeasures worldwide. In addition, the power of effective distances is that they vary in a time-dependent and environment-dependent manner, as they are influenced by the number of infected people in the mobile population and the number of individuals taking preventive measures, which is difficult to be captured from a movement data-based network. Therefore, it is useful to calculate effective distances and estimate the network structure of infection transmission based on the actual number of infected individuals.

In this context, the dynamics of infected individuals in Japan may be useful to estimate effective distances of viral transmission. In the past year and a half, although there has been no explosion of infections similar to the tens of thousands of new infections and thousands of deaths per day observed in the United States and Europe, there has been a constant rate of infections observed in Japan. The inability to impose legally binding restrictions on the movement of people and the opening of commercial facilities has made it difficult to completely control the number of infected people; although, the cooperative nature of the Japanese might prevent the explosion of infections [32–36]. In other words, in Japan, it is possible to observe a discrepancy between effective distances based on movement data and on the number of infected people, which is difficult in countries or regions that have experienced a rapid increase in infection and repeated lockdowns that completely restrict movement. In addition, Japan has had three peaks of infection, two declarations of a state of emergency (i.e., with moderate restrictions on the movement of people), and two stages of travel campaigns, during which the government encouraged domestic travel by subsidizing travel expenses, between 18 March 2020 and 12 March 2021, which have provided abundant data for describing changes in effective distances (Figure 1). Thus, the unique dynamics of SARS-CoV-2 infection in Japan is a suitable model for calculating the effective distance of the number of infected people and provides a new lockdown strategy based on an effective real-life-based network.

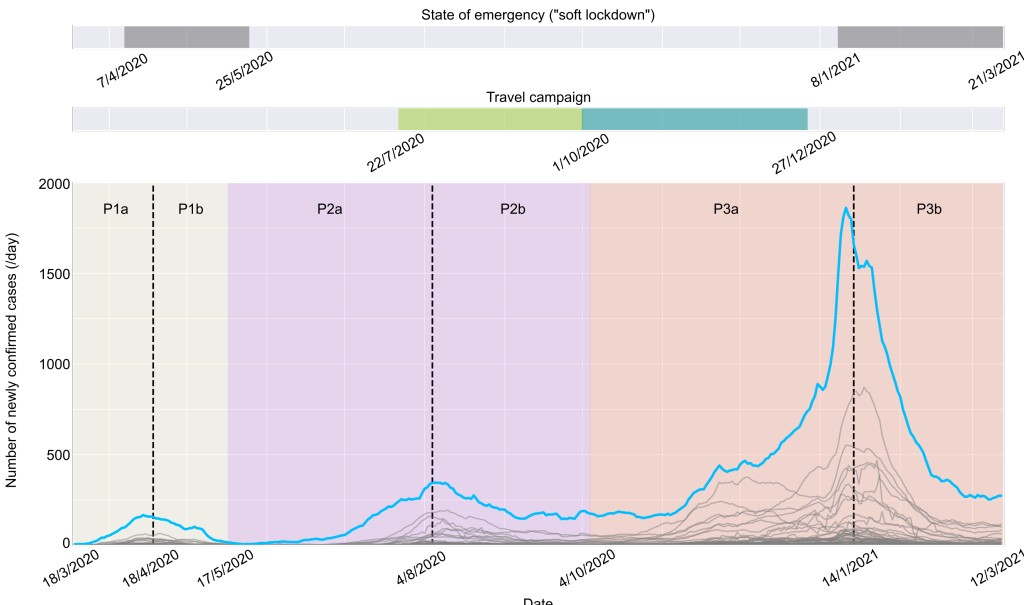

**Figure 1.** Timeline of the pandemic of coronavirus disease 2019 (COVID-19) in Japan. The period from 18 March 2020 to 12 March 2021 was considered. The top row indicates the period during which a state of emergency (i.e., "soft lockdown", or mild restrictions on movement, alcohol-serving businesses, and large events) was declared. The second row indicates the period of a travel campaign, "Go To Travel", during which the Japanese government encouraged the citizens to travel domestically by subsidizing travel expenses. The travel campaign had two stages: (1) the first from 22 July 2020 to 30 September 2020, during which Tokyo was excluded from the campaign, and (2) the second from 1 October 2020 to 27 December 2020, during which Tokyo was included and the subsidy was increased from stage 1. The main panel exhibits the 7-day backward moving average of the number of newly confirmed cases in each prefecture, which are highlighted in blue for Tokyo. Depending on the phase of the pandemic, the period was divided into six different periods.

In this study, we first fit the "Susceptible-Exposed-Infectious-Recovered" (SEIR) model to represent the dynamics of COVID-19 infection in Tokyo, the capital and the most populated city in Japan. We adopted the SEIR model because the model is not so complicated and can relatively easily accommodate other mathematical formulations to capture the dynamics of the COVID-19 pandemic better [37–42]. We then integrated diffusion equations, which formulate the diffusion process of random movements of particles, to model the spatio-temporal dynamics of COVID-19 across Japan and calculate the effective distance. We tested the effective distance model by envisioning two different declarations of a state of emergency and a travel campaign. Finally, a map of Japan based on effective distance was drawn, and this map was used to predict the dynamics of infection associated with the Tokyo 2020 Olympics.

## 3. Results

### 3.1. Diffusion of Infected Population from Tokyo Recapitulated the Propagation of COVID-19 Pandemic in Japan

To model the propagation of the COVID-19 pandemic in Japan as a diffusion process of the infected population from Tokyo and to estimate effective distances, we first recapitulated a time series using the SEIR model in which the number of infected people in Tokyo depended on the distinct phases of the pandemic, or periods P1, P2, and P3 (Figure 1). We calibrated the parameters for the number of potentially newly infected individuals in Tokyo during each period using the Markov chain Monte Carlo (MCMC) algorithm, and the summary of the estimated parameters is listed in Table S1, which were generally consistent with previous studies on COVID-19 [37,42]. Models using the estimated parameters accurately recapitulated the temporal dynamics and amplitude of the outbreaks and

surges in infections (Figure S1a–c). Leveraging these models as sources, we subsequently set up diffusion equations that were calculated as one-dimensional partial differential equations (PDEs) (see Section 5). Thus, human mobility outcomes differed and depended on whether the pandemic was on a spreading or shrinking trend; hence, we solved the diffusion equations dividing each phase into periods before and after a peak of infection in Tokyo, which generated six periods (i.e., P1a, P1b, P2a, P2b, P3a, and P3b) (Figure 1). After numerically solving the PDEs, we fit the number of newly confirmed cases in each of the 47 prefectures to the solutions and estimated the effective distance from Tokyo (Figures 2 and S2). The calculated results at the estimated effective distance, particularly during the first part of each period (i.e., P1a, P2a, and P3a), recapitulated the temporal evolution of the number of newly confirmed cases in most prefectures (Figure 2). Consequently, by considering the propagation of the pandemic as a diffusion process, we estimated the effective distance from Tokyo based on only the time series of newly confirmed cases of COVID-19 in each prefecture.

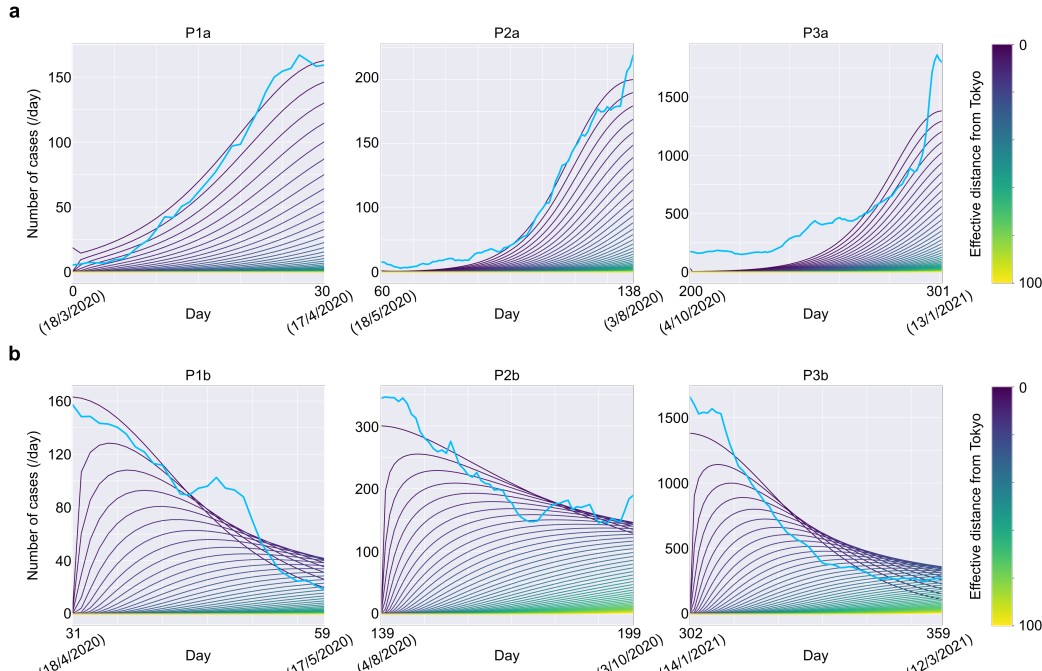

**Figure 2.** Effectivedistance from Tokyo is estimated from the diffusion process. (**a**,**b**) Solutions of diffusion equations for each period (**a**): before peaks of infection, (**b**): after peaks of infection. Solutions for $x \in \{2n \mid 0 \leq n \leq 50, \ n \in \mathbb{Z}\}$ over time are shown. The source of the diffusion process ($x = 0$) is the solution of the SEIR model, whose parameters were calibrated based on the MCMC algorithm (Figure S1a–c).

### 3.2. The Effective Distance Estimated Based on the Diffusion Process from Tokyo Revealed the Mobility of Individuals during the Pandemic

To characterize the estimated effective distance based on the diffusion process from Tokyo, we applied linear regression in which the geographical distances from Tokyo or effective distances based on inter-regional passenger traffic are explanatory variables [31]. As for the effective distance derived from passenger traffic data, we used inter-prefecture passenger volume data obtained in Japan in 2019, built a weighted directed network of mobility probability between prefectures, and estimated it as the shortest path from Tokyo in the graph. The effective distance based on the passenger traffic network generally predicted the effective distance based on the diffusion process, whereas the geographical distance did not, indicating that human mobility from Tokyo to other prefectures largely explained the time series of infections (Figures 3a and S3a–c). Notably, in the Kanto region, which consists of seven prefectures centered around Tokyo, both the distance metrics predicted the

effective distance derived from the diffusion equations with high accuracy, implying that, around Tokyo, the dynamics of infections could be largely explained by inter-prefecture travel between Tokyo and other prefectures, which correlated with the geographical distance on a local scale (Figures 3b and S3a). Conversely, in prefectures far away from Tokyo, there would be some factors other than the diffusion of infected individuals from Tokyo that contributed to the dynamics of infections. To examine the factors that contributed to the deviations of the effective distance based on the diffusion process from the predicted values in Figure 3a, we investigated residuals from the predicted values. The residuals of some regions exhibited unequal distributions around the baseline (Figure S4a,b). For example, residuals of the Kinki regions and the Tohoku regions are mostly distributed above and below the baseline, respectively (Figure S4a,b). These results imply that the deviations from the predicted values are attributed to local or regional factors other than travels from Tokyo, such as within-prefecture spread or containment of the coronavirus cases or travels from nearby prefectures. Therefore, our findings revealed that the effective distance based on the changes in the evolution of infection over time in each prefecture putatively reflected both human mobility and other local effects from nearby prefectures during the pandemic.

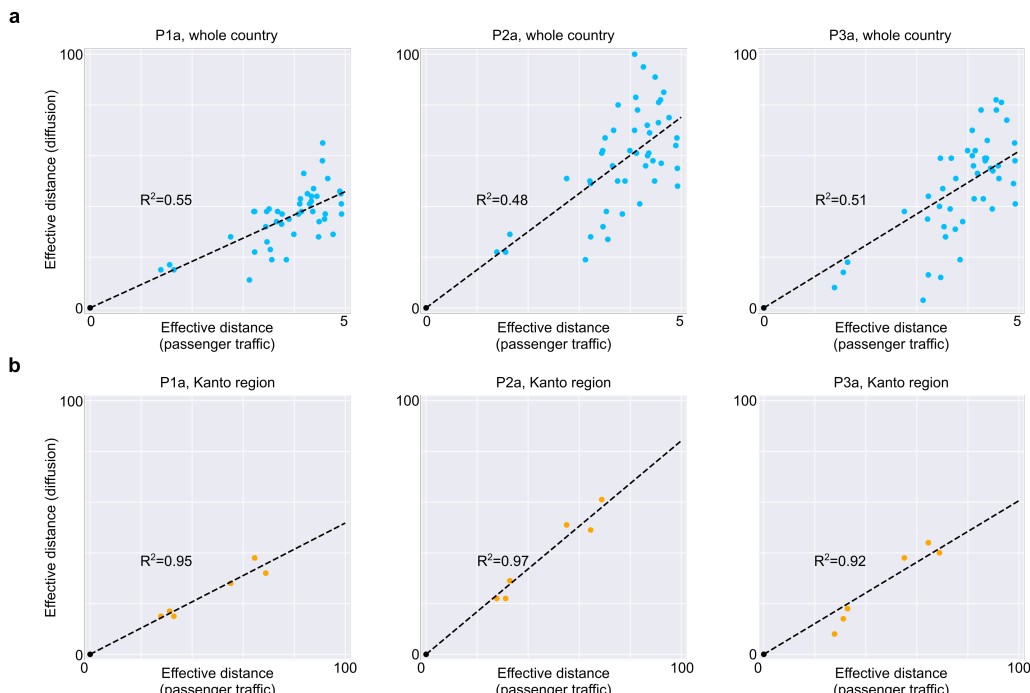

**Figure 3.** Effective distance derived from diffusion equations contains both passenger traffic and other local information that contribute to the spread of the pandemic. (**a**,**b**) Relationship between effective distance based on diffusion process from Tokyo and passenger traffic data in 2019 for all prefectures (**a**) and the prefectures in the Kanto region (**b**). A point represents each prefecture, and the prefectures whose effective distance is larger than 100 were excluded. Linear regression by least mean square and correlation coefficients are shown for each panel.

### 3.3. Effective Distance Changes Dynamically Depended on Stages of the Pandemic

Another advantage of the effective distance derived from time series data of infections over that derived from passenger volume data is that human mobility can be estimated dynamically without the need for extensive surveys. Such dynamic estimations based on different stages of the pandemic are particularly important to evaluate the effects of specific policies (e.g., "soft lockdown" and travel campaigns) and to define a better strategy. Thus, to investigate changes in human mobility over different stages of the pandemic, we first examined the distribution of the effective distance over the respective periods. Histograms of effective distances revealed that the shapes of the distributions shifted de-

pending on different periods, particularly during spreading periods P1a, P2a, and P3a (Figures 4a and S6a). Regarding the distribution of the effective distance, P2, during which the number of infected people declined without the need for any strict movement restrictions, our model showed that the largest mean and effective distance of each prefecture distributed widely around the mean (Figure 4a). Conversely, P1 exhibited the smallest mean and deviation from the mean, whereas P3 exhibited an intermediate mean, but the distribution was not unimodal, indicating the existence of distinct trends in human mobility between prefectures (Figure 4a). Considering that the Japanese government declared a state of emergency during P1 and P3 to contain the pandemic, the distribution of the effective distance of P2 would be considered desirable to limit the diffusion of the coronavirus pandemic while maintaining passenger mobility. Furthermore, during P3, the effective distance in some prefectures became smaller while others did not, presumably reflecting the increase in travelers from Tokyo to other prefectures due to the second stage travel campaign in which travels to and from Tokyo became subject to subsidization. These results demonstrated that infection-related human mobility could be captured during a pandemic. Subsequently, we investigated the changes in the effective distance for each prefecture by region. The general trend of the effective distance across the entire country was conserved over time (Figures 4b and S6b), which indicated that, other than diffusion from Tokyo, there was local connectivity between nearby prefectures that contributed to the spreading of the virus and altering their distance from Tokyo. Furthermore, considering the combination with other plots (Figures 4a and S5a,b), it can be implied that closer effective distances, in particular those found in major cities in each region, chiefly contribute to the rapid increase in coronavirus cases against which the government needed to impose restrictions. Consequently, the effective distance estimated from the time series of the newly confirmed cases evolved temporally, which might provide useful information for determining the appropriate mobility level during the pandemic.

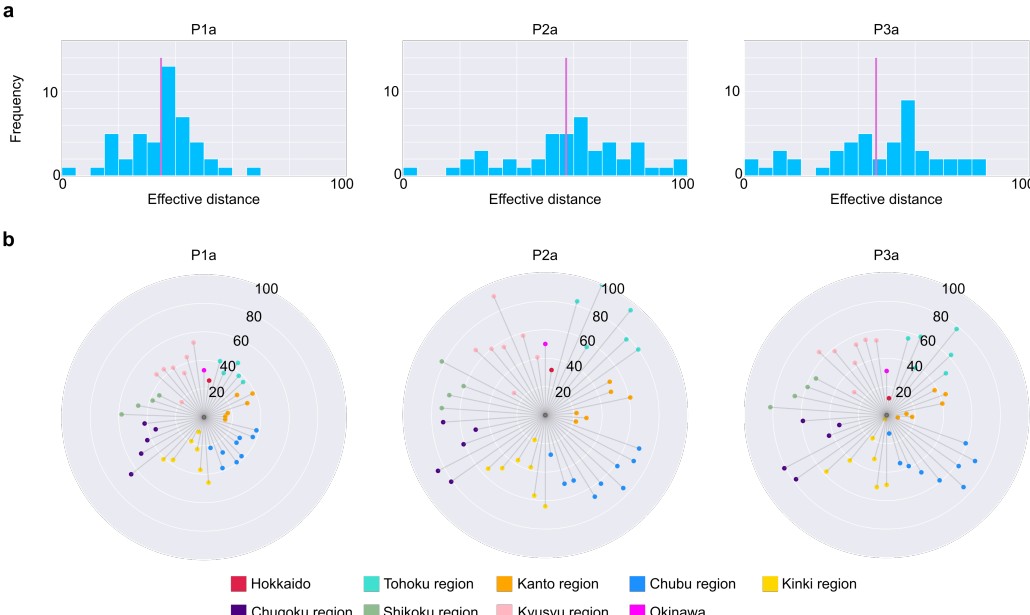

**Figure 4.** Effective distance changes dynamically depending on the stages of the pandemic. (**a**) Distribution of effective distances in periods P1a, P2a, and P3a. The vertical line in magenta indicates the mean value of the distribution. Prefectures whose effective distance is larger than 100 were excluded. (**b**) Effective distance of each prefecture in periods P1a, P2a, and P3a, ordered by "Prefecture Code" defined International Organization for Standardization. Regions to which prefectures belong are colored differently.

### 3.4. A Distorted Map of Japan, Based on Effective Distances and Local Interactions, Revealed the Non-Uniform Spread of the Pandemic across the Country

To visualize how the pandemic spread across Japan across different periods, we combined information on effective distances and local interactions between prefectures, which we defined as the geographical distance between adjacent prefectures that had large passenger traffic in the 2019 survey, to create a distorted map of the host country. The position (i.e., longitude and latitude) of a capital city in each prefecture was determined as the solution of a mathematical optimization equation using the non-linear least squares calculation [43]. In general, regions to the west of Tokyo, largely the Kinki and Kyusyu regions, reduced in size, whereas prefectures around Tokyo were expanded (for example, the Chubu regions were stretched in the northwest direction) in the distorted maps compared to the original one (Figure 5a–d). These deformations indicated that the coronavirus tended to rapidly spread to the western regions of the country, while it required additional time to spread to the north and north-western directions, particularly towards the Tohoku and Chubu regions. Thus, distorting maps based on effective distance and local interactions would be useful to characterize the spread of the coronavirus geographically and to develop specific policies aimed at preventing a large spike in the infectious population.

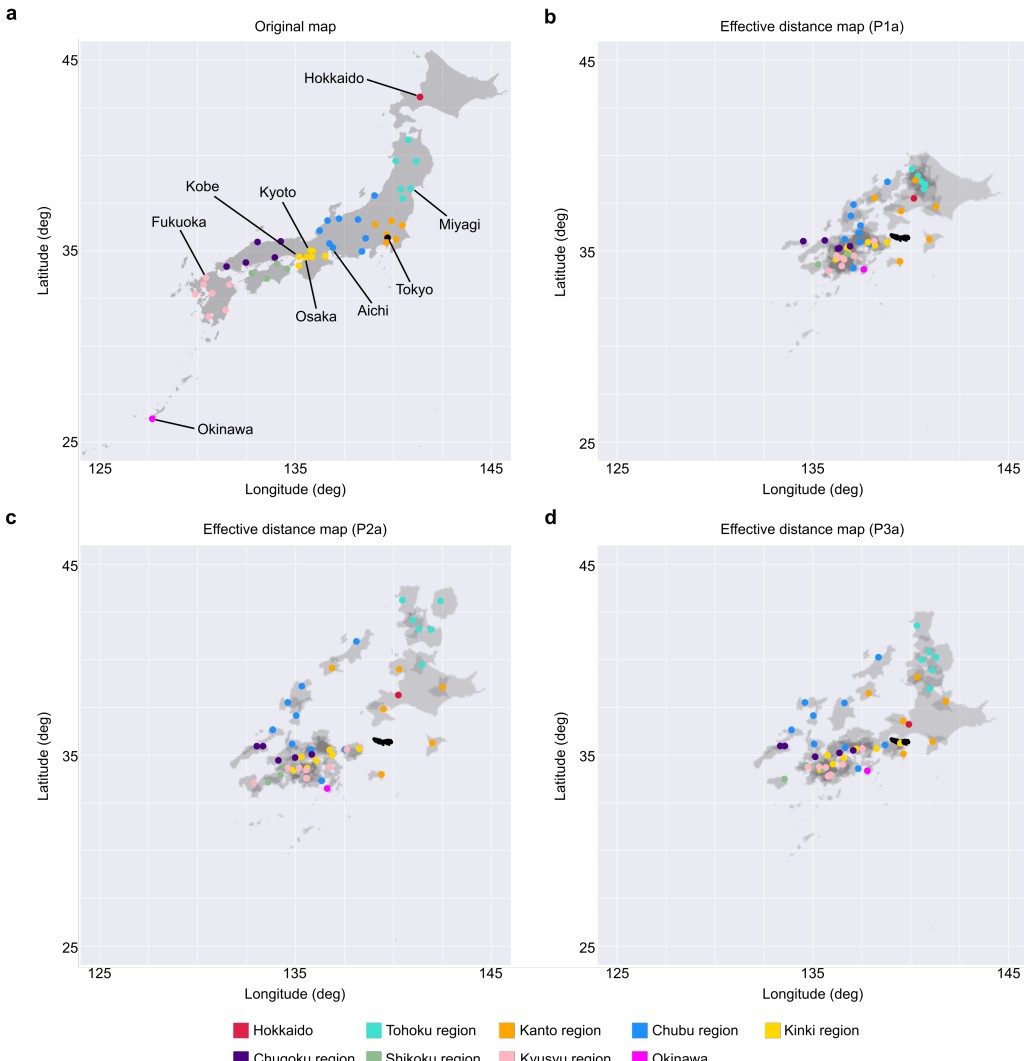

**Figure 5.** Distorted maps of Japan based on effective distance and local interactions reveal non-uniform spreading of the virus across Japan. (**a**) Original map of Japan. Each point indicates the capital of each prefecture, colored by region. (**b**–**d**) Distorted maps of Japan based on effective distance and local connectivity in periods P1a, P2a, and P3a (Figure S7).

### 3.5. Quantitative Estimation of the Scale of the Pandemic in Each Prefecture in Different Sets of Effective Distances

Although effective distance allows us to estimate how the virus spreads across the country, it is still unclear how much the effective distance contributes to determining the scale of the pandemic quantitatively, which would be a useful parameter when developing policies that balance both economic activity and medical resources in each prefecture. To evaluate the effects of the effective distance, we conducted simulations in which only the effective distance was changed from P3. If the effective distance was the same as P1, the total number of infected people would increase to more than twice in peripheral or rural areas, but would not increase to a similar degree in populated prefectures (Figure 6a). Conversely, if the effective distance was the same as P2, there would be no clear differences between the total number of infected individuals between the central or peripheral prefectures (Figure 6a). These results indicated that the effective distance could largely affect the scale of the pandemic, and these effects were heterogeneous between prefectures. Finally, we considered the increase in human mobility that could result from a very large public event such as the 2020 Olympic and the Paralympic Games, which were to be held in Tokyo and several other prefectures, and estimated the impact on the pandemic (Figure S9). Our findings showed that prefectures that held several matches, particularly prefectures that usually did not share strong connectivity with Tokyo, would experience a large increase in the number of patients (Figure 6b). For example, the number of infected cases in Fukushima prefecture would increase by 80%, while other host prefectures would increase by 40 to 60%. This indicated that the impact of an increase in human mobility due to huge public events was more severe in prefectures that usually had a larger effective distance from Tokyo. Based on these quantitative estimations, we recommend that policymakers should correctly control for the effective distance or distribute medical resources across the whole country. Therefore, effective distance enables us to estimate the scale of the pandemic in a single prefecture based on changes in human mobility and would be helpful in devising countermeasures against COVID-19 and other infectious diseases over changing situations.

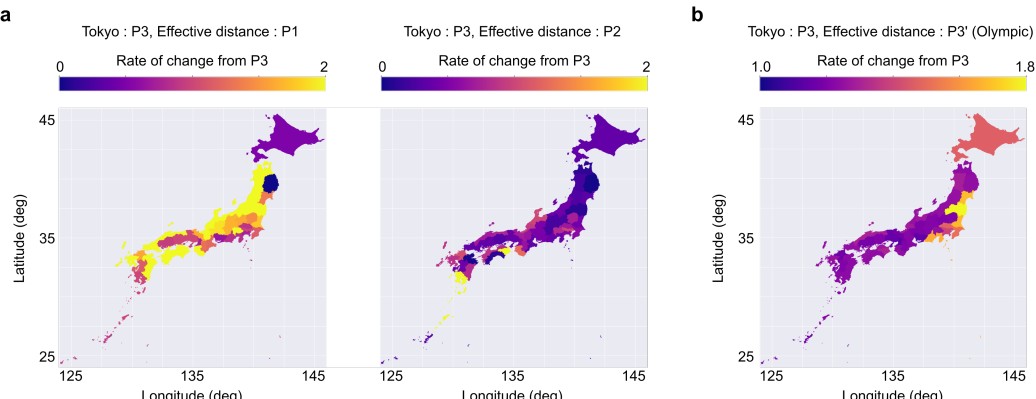

**Figure 6.** Simulations quantitatively estimating the effects of effective distance on a scale of the pandemic in each prefecture. (**a**) Estimated increase or decrease of total infection during an equivalent period as P3 when the effective distance from Tokyo is set to the same as P1 (left) or P2 (right) and the source (Tokyo) is the same as P3. (**b**) Estimated increase in infections of infection during an equivalent period as P3 when every prefecture approaches Tokyo because of a large-scale event such as the Olympics. Some prefectures where some games are held are closer to Tokyo than others. The results of the simulated shifts are shown in Figure S9.

## 4. Discussion

In this study, we calculated the effective distance of the spread of COVID-19 in Japan by applying the SEIR model to Tokyo, the capital city of Japan. Despite the presence of social factors that altered human mobility, such as two declarations of a state of emergency and travel campaigns, the spread of COVID-19 was demonstrated mainly by assuming

the spread from Tokyo during all phases of the pandemic (Figure S1). More specifically, our SEIR model-based predictions revealed that the effective distance from Tokyo to other prefectures changed over each phase, in which the effective distance between Tokyo and Hokkaido was close for P1 and P3, but far for P2, for example (Figure S1). Comparing the fitting parameter and effective distance in P1a, P2a, and P3a, we can see the effect of government-sponsored travel promotion. In particular, the effective distance between Tokyo and prefectures with many tourist attractions has increased. These findings provided an explanation for the spread of the two outbreaks in Hokkaido and many other prefectures, which could not be explained merely by passenger traffic data alone.

In Figure 6, we estimated the impact of a huge public event such as the Tokyo Olympics by creating an infection model that took into account the increase in the mobility of people between prefectures having event venues. Since the mobility of people between Tokyo and each prefecture was predicted to be a critical factor for pandemic control [24–26], the first priority for policymakers should be to prevent the infectious situation in Tokyo from being transferred to local areas, and in this sense, the decision of the Japanese government to hold the Olympic games without spectators is commendable. Conversely, although spectators have been excluded from participating in the events, athletes and media personnel remained a group of individuals moving around the event venues, so it is important to adequately consider the control strategy.

Estimating the effective distance from actual infection data is also a useful method for constructing an infection network for the spread of infection, constructing all infection networks based on the SEIR model, and simulating the spread of infection when an outbreak occurs in a city is computationally infeasible. However, once the network structure is drawn, a feasible simulation becomes available. For example, in the case of this study, by assuming the spread of infection from Tokyo, we were able to explain the infection situation in other prefectures with high accuracy (Figure S3), suggesting that the infection network in the country was a simple radial structure centered on Tokyo. Conversely, if the same method is applied to the United States or European countries, where multiple cities or countries, respectively, experienced surges of infection simultaneously [44,45], it may be possible to obtain a more complex structural organization of the infection network.

By combining the effective distance estimated from infection data with the effective distance based on a transportation network and movement data, we believe that it is possible to establish more effective preventive policies to avoid the spread of infection. It is difficult to quantitatively evaluate the contribution of various modes of transportation, such as airplanes, cars, and trains, to infectious diseases. By comparing the two effective distances, it is possible to determine the means of transportation that contribute to the transmission of the disease and to take selective and effective preventive measures. However, the proposed method has some limitations. Importantly, the assumption of the diffusion process from Tokyo is qualitatively different between periods before and after peaks of infection and does not allow us to quantitatively compare the effective distances between these periods. Thus, improvements in methodology to estimate the effective distance based on the uniform assumption remain to be addressed in future studies.

## 5. Methods

We confirm all relevant ethical guidelines and regulations regarding human data have been followed.

### 5.1. Dataset

We downloaded the data of the daily number of individuals who tested positive in 47 prefectures from the COVID-19 Situation Report in Japan by the Toyo Keizai Online COVID-19 Task Team [46]. The data were originally provided by the Ministry of Health, Labor, and Welfare. Returnees on government charter flights from Wuhan, airport quarantine and passengers, and cases on the *Diamond Princess* cruise ship were excluded. Due to the data availability from all of the prefectures, we took into consideration the data from 18 March

2020 to 13 March 2021, and divided it into three periods based on the three waves of the pandemic: (P1) from 18 March 2020 to 16 May 2020, (P2) from 17 May 2020 to 3 October 2020, and (P3) from 4 October 2020 to 12 March 2021. Each of these periods was further subdivided into two periods (a and b) depending on the time of the peak of infection. The timeline of the study and the major announcements from the Japanese government are illustrated in Figure 1.

The survey data on passenger traffic between prefectures in 2019 were from the Ministry of Land, Infrastructure, Transport, and Tourism. The data included the passenger volumes on railways, cars, ships, and airlines. The population data for each prefecture, the distance between prefectures, which is defined by the distance between prefectural capitals, and the GeoPackage for Japan were downloaded from the Statistics Bureau of Japan, the Geospatial Information Authority in Japan, and the Database of Global Administrative Areas (GADM), respectively.

### 5.2. SEIR Model of COVID-19 Pandemic in Tokyo

We consider the mathematical model in form of SEIR to describe the course of the COVID-19 pandemic in Tokyo. The choice of our model was motivated by several previously published studies on the COVID-19 pandemic [30,37,42].

The SEIR model consists of four compartments: Susceptible (S), Exposed (E), Infectious (I), and Recovered (R). The local dynamics of transmission in the model were given by

$$\dot{S} = -\beta I S$$
$$\dot{E} = \beta I S - \epsilon E$$
$$\dot{I} = \epsilon E - \rho I$$
$$\dot{R} = \rho I.$$

In the model, the susceptible population ($S$) becomes exposed to the viral agent upon contact with the infectious population ($I$) and thus, becomes the exposed population ($E$). The transmission rate of the virus is expressed as $\beta$, and $\beta I$ represents the force of infection. The exposed population becomes an infectious population at the rate $\epsilon$ (i.e., calculated as the inverse of the latent period), and the infectious population becomes the recovered population ($R$) at the rate $\rho$, the recovery rate (Table S1).

### 5.3. Parameter Estimation Using the MCMC Algorithm

Model parameters were estimated using the Bayesian framework by sampling the posterior parameter distribution via an affine-invariant MCMC algorithm using the emcee v3 toolkit [47,48]. For data reliability, the data used to calibrate the model consisted of the 7-day backward moving average of newly confirmed cases in Tokyo. Considering the rapid increase in the spread of the coronavirus pandemic in Tokyo, we assumed that the count data of newly confirmed cases followed a Gaussian distribution with a mean given by $\xi I$ and variance given by $1/\tau$. $E_0$ and $I_0$, the number of exposed and infectious individuals on the first day during the period, respectively, were also estimated. For simplicity, the number of recovered individuals, or $R_0$, was set at zero; thus, the number of susceptible individuals, or $S_0$, was calculated as $S_0 = N_t - E_0 - I_0$ where $N_t$ represented the total population of Tokyo. Consequently, we estimated seven parameters for each period: $\beta$, $\epsilon$, $\rho$, $\xi$, $E_0$, $I_0$, and $\tau$. We used the same uniform distributions for all periods as the prior distributions for each parameter, as described in Table S1. The statistics of the posterior distributions are also listed in Table S1. The convergence of the MCMC chains is judged based on the integrated autocorrelation time in accordance with a previous study [47]. When the number of samples satisfied the inequality given by,

$$M \geq C \hat{\tau}_f(M),$$

where $\hat{\tau}_f(M)$ was the estimated integrated autocorrelation time, the relative error in the estimation of the posterior distribution was considered sufficiently small.

### 5.4. Model of the Spread of the Infected Population from Tokyo and Estimation of Effective Distance

The spread of the COVID-19 pandemic in prefectures other than Tokyo was modeled as the diffusion process of infected individuals from Tokyo. We denoted the number of newly confirmed cases as $u(x, t)$, as follows:

$$\frac{\partial u}{\partial t} = D\frac{\partial^2 u}{\partial x^2}, \; x \in (0, \; L], \; t \in (0, \; T],$$

where $D$ was the diffusion coefficient. The parameters used in the simulation are summarized in Table S2. For simplicity, the initial conditions for the PDE were expressed as:

$$u(0, \; 0) = \xi I_0$$
$$u(x, \; 0) = 0, \; x \in (0, \; L],$$

which meant that the infectious population was located only in Tokyo in the initial condition. For the boundary conditions, we applied the Dirichlet boundary condition for $x = 0$ and the Neumann boundary condition for $x = L$, or

$$u(0, \; t) = \xi I$$
$$\left.\frac{\partial u}{\partial x}\right|_{x=L} = 0.$$

The parameters for the PDE are summarized in Table S2. The PDE was replaced by a series of ordinary differential equations (ODEs) by finite difference approximations, which were solved using the SciPy software. The solutions of the PDE, or the diffusion process of the infectious population from Tokyo depending on $x$ and $t$ values, are shown in Figure 2a,b.

To estimate the effective distance from Tokyo at different stages of the pandemic, we considered the diffusion process for six different periods depending on the period P1 to P3 and the peaks of the number of new cases in Tokyo during the periods: P1a, P1b, P2a, P2b, P3a, and P3b. The fitting of the observed data to the simulated data was also conducted for each period separately, and the $x$, or the effective distance of each prefecture from Tokyo based on the diffusion process, was determined by choosing $x$ with a minimum root mean square error (RMSE) value. The estimated effective distance for all periods are summarized in Table S3. If the effective distance of Tokyo, $x_T$, estimated from the fitting was not equal to zero, the effective distance of all prefectures was adjusted by subtracting $x_T$ for subsequent analyses.

### 5.5. Effective Distance Based on the Inter-Prefecture Network of Traffic and Mobility Data of Individuals

Since the contagion process is predominantly caused by the movement of people, we can also estimate the effective distance between prefectures based on passenger traffic data. The effective distance derived by this method was used to test the validity of the effective distance derived from the diffusion equation and for its characterization (Figures 3a,b and S3a). We defined the connectivity matrix **P** as follows:

$$P_{nm} = \frac{F_{nm}}{F_m} \; (0 \le P_{nm} \le 1)$$
$$F_m = \sum_n F_{nm},$$

where $F_{nm}$ represented the inter-prefecture passenger traffic on railways, cars, ships, and airlines. Thus, the matrix **P** quantified the fraction of the passenger flux from pre-

fecture $n$ to $m$. Given this flux-fraction matrix $\mathbf{P}$, the effective distance $d_{nm}$ from prefecture $n$ to $m$ is defined as

$$d_{nm} = (1 - \log P_{nm}) \geq 1.$$

We then constructed a weighted directed graph network consisting of 47 nodes, representing the prefectures in Japan, and edges from prefecture $n$ to $m$ with weights of $d_{nm}$ (i.e., $_{47}C_2 = 1081$ edges in total). Consequently, the effective distance from Tokyo to prefecture $n$, or $D_n$, was estimated as the length of the shortest path from Tokyo to the prefecture $n$ in the graph network.

### 5.6. Linear Regression of the Effective Distance Based on the Diffusion Process by Other Distance Metrics

The effective distance derived from the diffusion equations was predicted by a linear model without intercept, or $y = ax$, as Tokyo is at the origin. Models were fit to minimize the residual sum of squares. The coefficient of determination ($R^2$) was determined as $R^2 = 1 - \frac{u}{v}$, where $u$ is the residual sum of squares and $v$ is the total sum of squares.

### 5.7. Transformation of a Map of Japan Based on the Effective Distance from Tokyo and Local Connectivity

We created a transformed map of Japan according to the Mercator projection (i.e., the longitude and latitude corresponding to the $x$ and $y$ coordinates in the Euclidean plane, respectively) depending on the effective distance from Tokyo, which was estimated from the diffusion equation and local interactions between prefectures (Figure S7). Local interactions between prefectures were defined as the geographical distance between adjacent prefectures, including those connected by roads. Furthermore, for simplicity, the local connectivity was only considered for adjacent prefectures that had substantial passenger traffic or 100,000 people per year in at least one direction. The unit of effective distance was converted to that of geographical distance assuming a linear relationship between them locally, as shown in Figure S3c,d. In total, 111 links between prefectures (i.e., 46 links that expressed an effective distance from Tokyo to other prefectures and 65 links of local interactions) were used to transform a map of Japan to represent the spread of the pandemic.

Distorting a map based on a given proximity can be formulated as a nonlinear least squares problem as follows:

$$\min \sum_{ij \in L} \left( t_{ij} - \sqrt{(x_j - x_i)^2 + (y_j - y_i)^2} \right)^2,$$

where $ij$ denoted the links between points $i$ and $j$, and $x$ and $y$ represented the longitude and latitude of a prefecture, respectively. $L$ denoted the set of 111 links between prefectures, and $t_{ij}$ denoted the proximity (i.e., effective distance or local geographical distance). To solve this problem, we adopted the algorithm suggested by Shimizu and Inoue in 2009 [43]. In short, this algorithm considers the norm minimization of bearing changes instead of the norm minimization of variation of coordinates, which is solved by the Levenberg–Marquardt method, which uses bearings between points as the initial condition, and is a natural premise when transforming a map. Next, the nonlinear least squares problem was rewritten as

$$\min \sum_{ij \in L} \left[ \left\{ t_{ij} - (x_j - x_i) \sin \theta'_{ij} - (y_j - y_i) \cos \theta'_{ij} \right\}^2 \right.$$
$$\left. + \alpha \left\{ (x_j - x_i) \cos \theta'_{ij} - (y_j - y_i) \sin \theta'_{ij} \right\}^2 \right],$$

where $\theta'_{ij}$ was the approximate bearing of link $ij$ in the transformed map.

When $\alpha = 1$, we obtained:

$$\min \sum_{ij \in L} \left[ \left\{ t_{ij} \sin \theta'_{ij} - (x_j - x_i) \right\}^2 + \left\{ t_{ij} \cos \theta'_{ij} - (y_j - y_i) \right\}^2 \right].$$

This meant that we could approximate the solution for the initial nonlinear least squares problem by an iteration of solving two independent linear squares problems. When solving these linear squares problems, we assigned different weights to equations derived from the effective distance and those from local interactions (10:1). The weight was determined so that the distorted maps represented the effective distance from Tokyo as well as geographical relationships between nearby prefectures (Figure S8a,b). These weights enabled the effective distance from Tokyo to be expressed while retaining geographical information. When $|\theta_{ij} - \theta'_{ij}| < 0.01$, the solution was regarded as reaching convergence.

*5.8. Simulations to Estimate Effects of the Effective Distance on Scales of the Pandemic*

Period P3 was selected as the base period because we considered that the period reflected the most recent human mobility, and thus was suitable for estimating the impact of the Tokyo 2020 Olympic and Paralympic on the pandemic. The rate of change was calculated as the ratio of the estimated total number of infections in simulated distances to the total number of infections in the original distances (i.e., the effective distance during P3). The prefectures in which the 2020 Olympic and Paralympic Games were to be held (i.e., Tokyo, Kanagawa, Saitama, Chiba, Ibaraki, Shizuoka, Fukushima, Miyagi, and Hokkaido) were set three times closer than other prefectures, which modeled increased human mobility due to the games (Figure S9).

**Supplementary Materials:** The following supporting information can be downloaded at: https://www.mdpi.com/article/10.3390/app12189236/s1, Figure S1: SEIR model recapitulates time evolution of the number of newly confirmed cases in Tokyo for each period, Figure S2: Diffusion process from Tokyo recapitulates time series data of infections in other prefectures during period P1a, Figure S3: Relationship bewteen effective dis-tance based on diffusion and other distances, Figure S4: Residuals of the effective distance based on the diffusion process, Figure S5: Distribution of effective distance in different periods, Figure S6: Effective distance changes dynamically depending on stages of the pandemic, Figure S7: Local interactions with large passenger volumes, Figure S8: Distorted maps of Japan based on effective distance in different weight ratio between effective distance and local interactions, Figure S9: Putative map of Japan during the Tokyo 2020 Olympic used for the simulation in Figure 6b, Table S1: Parameters in the SEIR model and the estimated values, Table S2: Parameters in the diffusion equations, Table S3: Effective distance in each prefecture in the respective periods.

**Author Contributions:** T.Y. and S.S. designed the study, conducted the simulations, analyzed the data, and wrote the manuscript. All authors have read and agreed to the published version of the manuscript.

**Funding:** This work was supported in part by JSPS KAKENHI Grant—Grant-in-Aid for Early-Career Scientists, Transformative Research Areas A, and Moonshot R and D—MILLENNIA Program Grant.

**Institutional Review Board Statement:** Not applicable.

**Informed Consent Statement:** Not applicable.

**Data Availability Statement:** The raw data and source codes used in this manuscript are available at https://github.com/tyamadat/seir_pde, accessed on 3 August 2021.

**Acknowledgments:** This work was supported by JSPS KAKENHI Grant—Grant-in-Aid for Early-Career Scientists (19K16487 and 21K15136, S.S.), Transformative Research Areas A (20H05894 and 20H05903, S.S.), and Moonshot R and D—MILLENNIA Program Grant (JPMJMS2023-25, S.S.).

**Conflicts of Interest:** The authors declare no competing interests.

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
