# Peer review of "Estimating Infection-Related Human Mobility Networks Based on Time Series Data of COVID-19 Infection in Japan"

_applsci, doi:10.3390/app12189236_

Round 1
Reviewer 1 Report
Page 2:but various policies that have a great impact on human mobility have been developed. The author should give examples to these policies which might have effect on their study rather than social distance.
Page 5: Leveraging these models as sources, we subsequently set up diffusion equations. The author should declare what is this equations and its relevants inhere; as I do not get it; it is not clear the author should explain more about diffusion equation.
Page 5: Suplementary Table1: all suplementary tables should be rearranged; the title is given before each table; besides in Table 1 where is the relation between all these symbols used; it is not clear in the Table or in the text; I am confused.
The authors should write the number of infected people in each phase to make it easier to the reader.
Page 5: (dividing each period into periods before and after a peak of infection in Tokyo). I think it is better changed to (dividing each phase into periods.... ).
The authors should write the number of infected people in the whole study in each phase to make it easier to the reader.
Reviewer 2 Report
As your paper is on base of mathematical models so include the following papers for more interest and weight of the manuscript.
*Mathematical model for coronavirus disease 2019 (COVID-19) containing isolation class
*Dynamics of COVID-19 mathematical model with stochastic perturbation
*Dynamics of a fractional order mathematical model for COVID-19 epidemic
*Effect of weather on the spread of COVID-19 using eigenspace decomposition
*Deterministic and stochastic analysis of a COVID-19 spread model
*Crowding effects on the dynamics of COVID-19 mathematical model
Further, use once the word host country for Japan then throughout the paper use host country because repeated name has not a good impression over the readers. In model section amend the sentence "To describe the course of the COVID-19 pandemic in Tokyo, which is the capital and most populated city and likely the epicenter of the pandemic in Japan," with "We consider the mathematical model in form of SEIR to describe the course of the COVID-19 pandemic in Tokyo".
After these amendment I accept the paper for publication.
